# Ectopic Expression of Genotype 1 Hepatitis E Virus ORF4 Increases Genotype 3 HEV Viral Replication in Cell Culture

**DOI:** 10.3390/v13010075

**Published:** 2021-01-07

**Authors:** Kush K. Yadav, Patricia A. Boley, Zachary Fritts, Scott P. Kenney

**Affiliations:** 1Food Animal Health Research Program, Ohio Agricultural Research and Development Center (OARDC), The Ohio State University, Wooster, OH 44691, USA; yadav.94@osu.edu (K.K.Y.); boley.28@osu.edu (P.A.B.); 2Department of Electrical and Computer Engineering, University of Michigan, Ann Arbor, MI 48105, USA; zfritts@umich.edu

**Keywords:** hepatitis E virus, open reading frame 4, viral replication, genotype 1, cell culture

## Abstract

Hepatitis E virus (HEV) can account for up to a 30% mortality rate in pregnant women, with highest incidences reported for genotype 1 (gt1) HEV. Reasons contributing to adverse maternal-fetal outcome during pregnancy in HEV-infected pregnant women remain elusive in part due to the lack of a robust tissue culture model for some strains. Open reading frame (ORF4) was discovered overlapping ORF1 in gt1 HEV whose protein expression is regulated via an IRES-like RNA element. To experimentally determine whether gt3 HEV contains an ORF4-like gt1, gt1 and gt3 sequence comparisons were performed between the gt1 and the homologous gt3 sequence. To assess whether ORF4 protein could enhance gt3 replication, Huh7 cell lines constitutively expressing ORF4 were created and used to assess the replication of the Kernow-C1 gt3 and sar55 gt1 HEV. Virus stocks from transfected Huh7 cells with or without ORF4 were harvested and infectivity assessed via infection of HepG2/C3A cells. We also studied the replication of gt1 HEV in the ORF4-expressing tunicamycin-treated cell line. To directly show that HEV transcripts have productively replicated in the target cells, we assessed events at the single-cell level using indirect immunofluorescence and flow cytometry. Despite not naturally encoding ORF4, replication of gt3 HEV was enhanced by the presence of gt1 ORF4 protein. These results suggest that the function of ORF4 protein from gt1 HEV is transferrable, enhancing the replication of gt3 HEV. ORF4 may be utilized to enhance replication of difficult to propagate HEV genotypes in cell culture. IMPORTANCE: HEV is a leading cause of acute viral hepatitis (AVH) around the world. The virus is a threat to pregnant women, particularly during the second and third trimester of pregnancy. The factors enhancing virulence to pregnant populations are understudied. Additionally, field strains of HEV remain difficult to culture in vitro. ORF4 was recently discovered in gt1 HEV and is purported to play a role in pregnancy related pathology and enhanced replication. We present evidence that ORF4 protein provided in trans enhances the viral replication of gt3 HEV even though it does not encode ORF4 naturally in its genome. These data will aid in the development of cell lines capable of supporting replication of non-cell culture adapted HEV field strains, allowing viral titers sufficient for studying these strains in vitro. Furthermore, development of gt1/gt3 ORF4 chimeric virus may shed light on the role that ORF4 plays during pregnancy.

## 1. Introduction

Hepatitis E virus (HEV) is one of the most common causes of acute viral hepatitis and a major causative agent of acute fulminant hepatitis worldwide, with more than 20 million infections per year. HEV is responsible for chronic hepatitis in immunosuppressed patients but can also exhibit extrahepatic manifestations, such as neurological syndromes, renal injury, and hematological disorders [1,2,3]. *Orthohepevirus* A and C species of the *Hepeviridae* family are known to cause hepatitis E disease in humans [4,5]. HEV genotype 1 (gt1) and genotype 2 (gt2) are obligate to humans and mainly transmitted enterically, by drinking contaminated water, causing acute hepatitis in low-income and middle-income countries [6]. Zoonotic HEV genotypes 3 (gt3), 4 (gt4), and 7 (gt7) have been detected in both animals and humans, with pigs being the main reservoir for HEV gt3 and gt4 and camels for HEV gt7. The virus is transmitted by eating raw or undercooked infected meat, causing acute and chronic hepatitis [7,8,9,10,11]. Although typically self-limiting with a 2% mortality rate, the virus is highly detrimental to pregnant women during the third trimester, leading to a 30% mortality rate [12]. Ribavirin and IFN-α (PEGIFN-α) are contraindicated in pregnant women, thus limiting the therapeutic measures against HEV infection [13,14].

HEV is a positive-sense, 5′-capped, single-stranded RNA virus of approximately 7.2 kb in length (Figure 1A) [15,16,17]. HEV encodes three open reading frames (ORFs) (ORF1, ORF2 and ORF3) seen in all genotypes [18]. HEV gt1 ORF4 has recently been identified as a novel reading frame embedded entirely within ORF1 in a different reading frame. Transiently expressing ORF4 produces a 20 kDa molecular weight protein detectable by Western blotting of cellular lysates [19,20]. The expression of this ORF4 protein is regulated via an internal ribosome entry site (IRES)-like RNA element that is upregulated via cellular endoplasmic reticulum (ER) stress. ORF4 protein is rapidly turned over within cells as it possesses a proteasomal degradation signal [19]. However, loss of the ubiquitination site within a predicted intrinsically disordered region of the ORF4 protein (alteration of 50th leucine to proline) (Figure 1B) observed in seven sequences isolated from fulminant hepatic failure (FHF) and acute hepatitis patients suggests that viruses producing proteasome-resistant ORF4 may be a contributing factor to negative patient outcomes [19]. ORF4 is known to enhance the replication of gt1 HEV in Huh7 cells by increasing the activity of viral RNA-dependent RNA polymerase (RdRp), located at the terminal 3′ end of ORF1 in response to ER stress [19]. Contrary to this, HEV gt2 to gt4 are not thought to encode ORF4 [19].

Historically, HEV research has been hindered by the inability to culture the virus to adequate titers in cell lines, making it difficult to understand the growth, replication kinetics, and virus–host interactions, hindering antiviral and vaccine development [21,22]. Significant improvements to in vitro virus culture occurred when a gt3 HEV, Kernow-C1 strain, isolated from a chronic HEV patient and serially passaged six times (P6) reported enhanced viral fitness in vitro. Sequencing revealed that insertion of a fragment of the human 40S ribosomal protein S17 in the ORF1 region endowed enhanced replication ability and broadened the host range of Kernow-C1 P6 [23,24]. Furthermore, S17-inserted gt1 HEV Sar55, when transfected to Huh7 S10-3 cells, did not acquire the cell culture-adaptive characteristics [25]. Many differing cell lines and strains have been utilized in the past to develop acceptable culture systems, but HEV replication progressed very slowly and infection with low virion counts resulted in non-productive infection [26,27,28]. Recent breakthroughs utilizing placental (JEG-3) cells, primary small intestinal cells, tissue explants, and stromal cells have been achieved demonstrating susceptibility to specific strains of HEV [29,30,31]. Hence, a range of different expression systems and cell lines have been used to study HEV without the need for authentic virus replication.

In this study, we sought to validate previous findings regarding ORF4, by determining whether gt3 HEV contained a similar ORF4 to that of gt1 HEV and whether ORF4-mediated replication enhancement could be bestowed upon other non-gt1 HEV genotypes by providing ORF4 in trans and observing the replication of gt3 HEV. Furthermore, we studied gt1 HEV replication in exogenously expressed ORF4+ Huh7 cell lines. We observed efficient replication and enhanced infectious particle production of different HEV strains used in the study.

## 2. Results

### 2.1. Kernow-C1 Strains (HEV gt3) Lacks the IRES-Like Element RNA Function and Start Methionine Associated with ORF4 Translation

An IRES-like RNA element is present in gt1 HEV from 2701 to 2787 bases (Sar55; GenBank accession number AF444002) [19,41]. Sequence alignment of the gt1 HEV IRES region with the homologous gt3 HEV sequence revealed some sequence conservation with only 24.13% (21/87) nucleotide differences within the IRES-like element (Figure 2A). To confirm the presence of the ORF4 IRES-like element in gt1 HEV and to empirically determine whether homologous gt3 HEV sequence could function as an IRES-like element, the IRES-like element sequences were inserted into a dual luciferase expression vector between sequences encoding firefly luciferase (FFL) and nano luciferase (NL). These dual luciferase IRES-like element vectors were then transiently transduced into Huh7 S10-3 cells and then treated with tunicamycin to induce ER stress. An increase in the NL ratio to FFL suggests the activation of the IRES-like RNA element which is known to upregulate ORF4 synthesis in gt1 HEV. Significant differences were observed between untreated gt1 and tunicamycin-treated gt1HEV for the relative ratio of NL to FFL in the presence of tunicamycin (*p* < 0.01) (Figure 2B). In addition, the initiator amino acid methionine for ORF4 is only present in gt1 HEV (Figure 2C). However, there are methionines in the downstream sequence in gt3 HEV (data not shown) but because gt3 HEV does not retain IRES-like element functionality, it does not possess the coding capacity to generate ORF4.

### 2.2. Lentiviral Transduction of Huh7 S10-3 Cells to Express HEV gt1 ORF4

To empirically test whether gt1 ORF4 can influence replication of non-gt1 HEV strains, we created a cell line constitutively expressing ORF4 from gt1 HEV. Lentiviral transduction utilizing the pULTRA packaging vector either empty or encoding gt1 ORF4 was used to create lentiviral particles pseudotyped with the vesicular stomatitis virus envelope glycoprotein (VSV-G) and subsequently used to infect the Huh7 S10-3 cell line (Figure 3A,B) [42]. GFP-positive Huh7 S10-3 cells indicated successful transduction via lentiviral infection (Figure 3C). Clonal cell lines were established via limiting dilution and selection of single GFP-positive cell colonies was done. Detection of the ORF4 mRNA in ORF4+ and ORF4− lentivirus transduced Huh7 cell lines was confirmed by RT-PCR (Figure 3D) on RNA samples that were DNase treated and confirmed free of DNA via a PCR reaction lacking reverse transcriptase (data not shown). ORF4 protein expression was confirmed via Western blotting of lentiviral transduced Huh7 S10-3 cells using a rabbit polyclonal antibody raised against recombinant 6×His-tagged ORF4 generated in *E. coli* (Figure 3E).

### 2.3. Enhanced Viral Replication of HEV gt3 (P1 and P6) and HEV gt1 in the Presence of gt1 HEV ORF4 at 5 and 7 Days Post-Transfection (dpt)

Although P1 and P6 strains belong to HEV gt3, there were marked differences in the ability to be expressed within Huh7 S10-3 cells [24,43]. This Kernow-C1 gt3 strain originated as a quasispecies in a chronically infected human patient and gained an in-frame insertion of 58 amino acids of the ribosomal protein S17 (RPS17) within the hypervariable region (HVR) of ORF1 [24]. Analysis of the predominant cDNA derived from passage 1 (P1) Kernow-C1 HEV (Gen Bank accession no. JQ679014) revealed that it lacked the RPS17 insertion and had 25 additional amino acid differences than cDNA from passage 6 (P6) Kernow-C1 strain (GenBank accession no. JQ679013). To determine whether gt1 HEV ORF4 protein could alter replication of HEV strains not naturally encoding ORF4, we utilized the Kernow-P1/P6 HEV cell culture system in conjunction with Huh7 S10-3 cells transiently expressing ORF4. To determine whether ORF4 could enhance viral replication, full-length viral RNA from HEV gt3 (Kernow P1, P6) and HEV gt1 (Sar55) was transfected into either empty vector transduced Huh7 S10-3 cells, ORF4+ expressing Huh7 S10-3 cells, and Huh7 S10-3 cells (Figure 4A). Furthermore, to study the effects of ORF4 provided transiently at 5 dpt, cells were fixed and probed for ORF2 expression via an immunofluorescence assay. As ORF2 is translated from subgenomic mRNA, it is produced only during the later stages of HEV replication and thus, serves as an indicator of complete active viral replication [36,44]. To directly show that HEV transcripts have productively replicated in the target cells, we assessed events at the single-cell level using indirect immunofluorescence and flow cytometry. Employing antibodies directed against ORF2 capsid protein, we detected cells expressing the ORF2 protein that suggests efficient replication (Figure 4A,B). Total numbers of ORF2 expressing cells were quantified by staining ORF2 encoded capsid antigen and with flow cytometer readings. As depicted in Figure 4C,D, gt3 HEV (P1 and P6) and gt1 HEV (Sar55) replication significantly increases when ORF4 is provided in trans. Higher percentage of ORF2-positive cells are observed with HEV gt3 P6 at both 5 and 7 dpt. Approximately, 25%, 8%, and 12% ORF2-positive cells are observed in HEV gt3 P6, P1, and HEV gt1 Sar55 transfected ORF4+ Huh7 cell lines at 5 dpt, respectively, compared to 13%, 6%, and 7% ORF2-positive cells of respective strains at 7 dpt. Hence, there is the marked reduction in the percentage of ORF2-positive cells from 5 to 7 dpt, suggesting either a decrease in ORF2 expression with time or cellular replication exceeds new viral infection. No difference between ORF4− Huh7 cell line and Huh7 cell line transfected with any of the HEV transcripts (P6, P1, and Sar55) suggests that the effects are specific to the presence of ORF4. In summary, these results demonstrate that ORF4− expressing cells supported enhanced viral replication.

### 2.4. ORF4-Enhanced Viral Replication of HEV gt3 (P1 and P6) and HEV gt1 (Sar55) Lead to Increased Numbers of Infectious Particles

To test whether enhanced viral replication led to an increase in the number of infectious HEV virions, we utilized HepG2/C3A cells for infection studies. Hence, we tested the ability of the HEV gt3 (P6 and P1) and HEV gt1 (Sar55) transfected target cell lysates with and without ORF4 expression to produce infectious particles (Figure 5A). To quantify, HEV ORF2 antigen expression in the infected cells, ORF2-positive cells were screened by flow cytometry. As shown in Figure 5B,C, higher numbers of infectious particles were seen with cellular lysates from HEV gt3 (P6 and P1) and HEV gt1 (Sar55) transfected ORF4+ Huh7 cell lines when compared to ORF4− Huh7 and Huh7 S10-3 cell lines. Approximately 10%, 5%, and 3.5% ORF2-positive HepG2/C3A cells were seen with HEV gt3 P6, P1, and HEV gt1 Sar55 transfected ORF4+ Huh7 cell lysates at 5 dpt used, respectively, compared to 5%, 2.5%, and 2.2% ORF2-positive cells of respective transfected cell lysates at 7dpt. No significant difference between the numbers of infectious particles from lysates of ORF4− Huh7 cell lines and Huh7 cell lines for all strains used in the study suggests that the results are attributable to ORF4 expression. Collectively, HEVgt3 (P6 and P1) and HEV gt1 transfected ORF4+ Huh7 cell lysates contain comparatively more infectious particles than lysates from ORF4− cellular lysates.

### 2.5. Enhanced Viral Replication of gt1 HEV (Sar55) in ORF4+ Expressing Huh7 S10-3 Cell Line in Presence of ER Stressor

Because transfection of HEV gt1 in cell culture does not yield high replication, we performed transfection of Sar55 gt1 HEV in the presence of ER stressor (tunicamycin) in ORF4-expressing cell lines. Further controls included HEV gt3 P6 RNA transcripts transfection of the ORF4+ Huh7 cell line. We assessed events at the single-cell level using immunofluorescence and flow cytometry. Employing antibodies directed against ORF2 capsid protein, we detected cells expressing the ORF2 protein that suggests efficient replication. Higher numbers of fluorescently staining cells were seen with HEV gt1 (Sar55) transfected in tunicamycin-treated cells in comparison to untreated cells (Figure 6A). Higher numbers of ORF2-positive cells were observed by flow analysis in tunicamycin-treated ORF4+ Huh7 cells than untreated cells (Figure 6B). In contrast, there was no significant difference seen in the number of ORF2-positive cells between tunicamycin-treated and -untreated ORF4+ Huh7 cell lines transfected with HEV gt3 P6. These results also suggest the lack of IRES-like element activity in HEV gt3 P6, thus suggesting no effect of tunicamycin in gt3 P6 replication. Together, gt1 HEV replication is further enhanced in ORF4+ Huh7 cell lines treated with tunicamycin.

## 3. Discussion

Historically, lack of an efficient in vitro culture system has limited HEV research [22]. Although the disease was discovered in the late 1980s, the reason behind higher pregnancy mortality has not been fully explained on a molecular level due to low propagation of virus in cell culture and failure of genotype 3 HEV to replicate pregnancy mortality in the pig model [43,45]. Recently, many groups have successfully propagated various HEV strains in non-hepatic cell lineages (e.g., A549, JEG-3, primary endothelial cells) but a highly propagating, efficient in vitro system of gt1 strains in human liver cells is still not available [29,30,31,46]. Lack of an efficient culture system and gt1 animal pregnancy model system is a major reason for the lack of knowledge regarding hepatitis E produced pregnancy mortality [47].

HEV gt1 is known to encode an additional reading frame (ORF4) only during ER stress. ORF4 is thought to play an important functional role in the viral replication cycle of gt1 HEV. ORF4 interacts with several viral proteins, promotes the assembly of the viral replication complex, and enhances the viral RdRp (RNA-dependent RNA polymerase) activity in the case of gt1 HEV [19,20]. In particular, ORF4 appears to enhance HEV replication on the level of transcription/translation by binding to and forming a multiple protein replication complex consisting of viral helicase, RdRp, the X domain, host Eef1α1 (eukaryotic elongation factor 1 isoform-1) and tubulinβ [19]. Interestingly, the gt3 RdRp, helicase, and the X domain have also been proposed to form a similar viral replication complex without the necessity of ORF4. Additionally, ORF3 was shown to inhibit this complex by competing with the X domain for the same binding region as ORF4. Sequence comparisons of the IRES-like regions and dual luciferase assay results confirm an absence of ORF4 in gt3, suggesting gt3 HEV has other factors responsible for enhanced replication in mammalian cells that have yet to be discovered or that ORF4 is dispensable for replication complex formation. Thus, knockout of ORF4 expression in the context of gt1 virus to observe replication complex formation and overall pathogenicity of the virus could provide additional mechanistic insights.

Our study addresses the ability of ORF4 to enhance replication across different genotypes of HEV. We show for the first time that gt3 strains of HEV, Kernow-C1 P1 (non-cell culture adapted) and P6 (cell culture adapted) can use the ORF4 protein from gt1 to enhance its replication in vitro. Interestingly, ORF4 is encoded only in gt1 HEV and bioinformatics analysis did not predict ORF4 in other genotypes of HEV [19]. Despite gt3 Kernow strains not encoding the ORF4 within its genome, the functionality of ORF4 protein to enhance replication remains. Additionally, Kernow P1 is non-cell culture adapted but in the presence of ORF4, there is a significant increase in the replication efficiency of the virus demonstrating its ability to be cultured more efficiently in ORF4+ expressing Huh7 cell line. Furthermore, gt1 HEV has been difficult to propagate in cell culture while exogenously provided ORF4 increased its replication. This finding suggests that expression of ORF4 can be utilized to boost replication of field strains of HEV that otherwise may not be able to be studied in vitro.

ORF4 is degraded quickly by the host proteasome and therefore is short lived [19]. Thus, it appears that gt1 HEV undergoes less efficient viral replication than gt3 HEV under normal cellular conditions but has adapted a tightly regulated replication enhancement during ER stress conditions. The reason for this mechanism has yet to be elucidated. However, there exists reports of multiple viral and cellular IRESs that are activated only under certain conditions. Encephalomyocarditis and foot-and-mouth disease virus IRES-mediated translation is regulated by 4E-BP1 under amino acid starvation [48]. Human cytomegalovirus latency protein pµL138 is translated by an IRES-like element during serum stress, and an IRES within human immunodeficiency virus-1 mediates viral structural protein synthesis during G2/M phase of cell cycle, under oxidative stress [49,50]. Hepatitis C virus IRES-mediated translation is controlled by phosphorylation of eIF2α under oxidative stress [51]. In addition, IRES elements enable virus translation to proceed in the absence of normal host translation mechanisms. Hence, we can speculate that under certain conditions of cellular stress when most translational activities are shut off, translation by IRES-like elements is differentially upregulated. This evolutionary adaptation is selectively advantageous to the life cycle and survival of the viruses and is essential for the regulation of specific genes involved in cellular processes such as survival and differentiation [52,53]. Similarly, we can speculate that during ER stress most of the other translational activity of host cell is shut off but gt1 HEV is still able to translate ORF4 protein by the help of the IRES-like element, aiding in genome replication.

ORF4 has been shown to enhance RdRp activity through the viral replication complex. In our study, Sar55 replication was enhanced in the presence of lentiviral transduced ORF4 by induction of ER stress. We speculate that limiting secondary factors in addition to ORF4 control ORF4 HEV mediated replication, or a second pathway may exist where ORF4 facilitates gt1 HEV to propagate well in cell culture during ER stress. Hepatotropic viruses such as hepatitis B are known to maintain HBV antigen expression and replication when HBx (non-structural protein) is constantly available. HBx stimulates HBV replication by activating viral transcription, initiating, and maintaining transcription and enhancing viral polymerase activity [54]. We speculate that ORF4 also uses other pathways which involve physiological importance to the HEV viral replication such as HBx for HBV. Further studies need to be done focusing on the pathways demonstrating how gt1 HEV uses ORF4 for higher replication leading to enhanced pathology. This may help elucidate the mechanism behind gt1 HEV strains virulence to pregnant women during the second and third trimester.

ORF2 is known to exist in several forms such as secreted form (ORF2^S^) and capsid form (ORF2^C^) and they are two different translation products of the viral ORF2 gene. Importantly, HEV infected cells release a large amount of non-virion associated ORF2 both in vitro and in vivo [37]. Previous studies demonstrated a decrease in ORF2 in cell culture reflecting a similar trend in the ORF4+ expressing cells from the 5th to 7th dpt in our study. A similar pattern was observed with the infected HepG2/C3A cells at 5th and 7th dpi. ORF2^S^ is not essential for the HEV life cycle in cultured cells, but it is able to reduce antibody-mediated neutralization in vivo [37]. In addition, it is known that HEV ORF2 inhibits cellular NF-kB activity by blocking ubiquitination mediated proteasomal degradation of cellular I kappa B alpha (IkBα) and inhibits the RIG-I mediated interferon response, thus evading host innate immune responses [55,56]. A possible explanation for enhanced replication may be increased ORF2 protein limiting RIG-I- mediated effects that typically reduce HEV replication.

In addition, the host immune system in response to viral infection secretes multiple compounds activating the stress response. The ER is essential for protein homeostasis and pathological stressors cause accumulations of misfolded proteins in the ER. The ER stress-induced unfolded protein response (UPR) restores ER homeostasis through dynamic intracellular signaling pathways but is associated with various diseases in which these cell-specific signaling functions fail to restore homeostasis [57]. Tight control of HEV replication via proteasomal degradation of ORF4 might be serving as an antiviral strategy by limiting gt1 HEV replication under normal cellular conditions to evade host detection. Once ER stress is triggered by viral detection the virus rapidly increases replication to generate as many infectious particles as possible prior to the infected cell being eliminated. Furthermore, HEV variants containing proteasome-resistant ORF4 (Figure 1B), preventing rapid turnover, have been shown in clinical cases with poor health outcomes likely due to increased viral production [19]. Further study regarding proteasome-resistant ORF4 and its effect upon disease outcome is warranted. Currently, the only animal model available that recapitulates pregnancy mortality is the rabbit utilizing gt3 [58]. The most well studied HEV animal model, the pig, is reliant on gt3 HEV strains which lack ORF4 expression. Testing HEV in the pregnant swine animal model did not reproduce enhanced mortality [43]. Creating a gt3 HEV which expresses gt1 ORF4 would allow us to evaluate the potential pathology associated with ORF4-mediated viral replication enhancement in vivo. The effects of the ORF4 protein in the replication enhancement of the gt3 HEV (P1 and P6) suggests the feasibility of developing a gt1/gt3 chimeric virus which could help in the development of the first pig model system for evaluating the role of ORF4 during HEV infection during pregnancy. The ability of gt3 HEV to utilize ORF4 strongly suggests that we can establish a gt3 HEV chimera that encodes gt1 ORF4. If viable, such a virus would allow us to assign definitive roles of ORF4 to pathogenesis. The pig model system could be used to understand the pregnancy related pathology and the contributing pro-viral factors contributing to fulminant hepatitis leading to a 30% pregnancy mortality rate in humans.

Several studies utilizing hepatocyte-like cells (HLCs) differentiated from iPSC/hESC (induced pluripotent stem cells/human embryonic stem cells) as an alternative cell culture model to hepatoma cells support intracellular HEV replication but not infection [26]. When endodermal cells differentiated to immature hepatocytes, they do become susceptible to HEV infection. However, cell culture non-adapted Kernow-C1 P1 strain replicated better than the adapted P6 strain in HLCs, suggesting that the acquired RPS17 mutations in cell culture attenuate viral replication in more physiologically relevant systems [26]. Similarly, ORF4−expressing Huh7 cell lines are not physiologically relevant but the cell culture-adapted gt3 P6 strain replicated better than the non-cell culture-adapted gt3 P1 strain. Hence, ORF4+ expressing Huh7 cells provide a reproducible platform to study genotype heterogeneity, viral or cellular determinants defining host range. Technology such as CRISPR/cas9 reflected that cyclophilin inhibits cell culture-adapted P6 but not the original Kernow-C1 P1 using HLCs [26]. We predict that our ORF4−expressing Huh7 cell lines with advanced CRISPR/cas9 technology could be used to evaluate new compounds that inhibit HEV replication. Furthermore, HEV has been demonstrated to complete the full viral life cycle in JEG-3 cells [29]. Hence, it would be interesting to study the replication of gt3 HEV in ORF4−expressing JEG-3 cell lines.

In conclusion, this improved HEV cell culture system provides a powerful tool to explore the ability to enhance the growth of non-cell culture adapted HEV field strains and warrants further study of the ORF4 protein in the presence of various pregnancy hormones. Future studies of differing HEV genotypes in cell systems such as ORF4−expressing Huh7 cells will significantly advance our understanding of HEV biology, help in the discovery of antiviral drugs and vaccines, and help to study host and viral factors producing HEV pathology in pregnant women.

## 4. Materials and Methods

### 4.1. HEV Infectious cDNA Clones, Cell Lines and Culture, Lentiviral Vectors, Lentiviral Transduction

Passage 1 (P1, lacking the RPS17 insertion, non-cell adapted) and passage 6 (P6, containing the RPS17 insertion, cell adapted) strains of gt3 Kernow-C1 HEV infectious cDNA clones were used [24]. The dual luciferase reporter plasmid was created by inserting a synthetic DNA construct encoding the coding sequences of firefly luciferase, the encephalomyocarditis virus (EMCV) IRES sequence, and coding sequence for nano luciferase into the pCDNA3.1 vector (Thermofisher, Carlsbad, CA, USA) using BamHI and XhoI restriction sites (A kind gift from Nicholas Catanzaro and XJ Meng, Virginia Tech, Blacksburg, VA, USA). Subsequent cloning utilized EcoRV and EcoRI restriction sites flanking the EMCV IRES coding sequence. The ORF4 IRES-like element sequence from gt1 was artificially synthesized based on the gt1 (Sar55 IRES, 2701–2787, GenBank accession no. AF444002) and the homologous IRES-like element region from gt3 (Kernow P6, GenBank accession no. JQ679013) using gBlocks (IDT, Coralville, IA, USA) gene fragments. For establishment of ORF4-expressing cell lines via lentiviral transduction, the ORF4 coding sequence was synthesized based upon Sar55 gt1 HEV sequence using gBlocks (IDT) gene fragments and inserted into the multiple cloning site of the pUltra packaging vector (Addgene plasmid, Watertown, MA, USA, #24129) using restriction site BamHI [42]. The pUltra vectors were co-transfected with packaging plasmid psPAX2 (Addgene plasmid #12260) and vesicular stomatitis virus G protein plasmid pMD2.G (Addgene plasmid #12259) into HEK293T cells using lipofectamine LTX (Invitrogen, Carlsbad, CA, USA). Forty-eight hours post-transfection, supernatants were harvested and used to infect Huh7 S10-3 cells on 6 well dishes. Huh7 S10-3 subclone (human hepatoma cells) was used for the study [59,60]. Limiting dilution method was used to prepare clonal cell lines [61,62] of ORF4+ (GFP plus ORF4) as clones 3, 4, 5 and ORF4− (only GFP) as clone 4.

### 4.2. Dual Luciferase Assay, RT-PCR, Gel Electrophoresis and Western Blot

Plasmids encoding gt1 IRES-like element sequence and homologous sequence from gt1 HEV were transfected into Huh7 S10-3 cells using lipofectamine LTX (Invitrogen). Twenty-four hours post-transfection, cells were either treated with 10 µg/mL tunicamycin (Sigma Aldrich, St. Louis, MO, USA) or DMSO vehicle control for 16 h. Cells were lysed, and luminescence measured using the Nano-Glo dual-luciferase reporter assay system (Promega, Madison, WI, USA) read on a Filtermax F5 plate reader (Molecular Devices, Sunnyvale, CA, USA). For PCR detection of ORF4 mRNA, total cellular RNA was extracted using Trizol reagent and treated with Turbo DNase (Fisher Scientific, Hampton, NH, USA) using the high stringency protocol. RT-PCR was performed and the ORF4 sequence was amplified using the primers ORF4 f 477 (5′-ATGTTGCGCGGACAGC-3′) and ORF4 rev 477 (5′-TTAGCTCACATACATCCGCAG-3′). Agarose gel electrophoresis was used to separate DNA and ethidium bromide staining was utilized to observe the presence or absence of ORF4 mRNA in ORF4+ and ORF4− cell lines, respectively. For protein analysis, the Huh7 S10-3 cells (ORF4+ and ORF4−) were seeded in 6 well dishes at density of 4 × 10^5^ cells/well for 24 h. Total cellular lysate was made utilizing 1.5× laemmli buffer (heated to 100 °C). A volume of 50 µL of lysate was subjected to sodium dodecyl sulfate polyacrylamide gel electrophoresis (SDS-PAGE) separation using mini-PROTEAN TGX stain-free precast gels, 4–20% (Bio-Rad, Hercules, CA, USA), followed by transfer onto a nitrocellulose membrane (Trans-Blot Turbo; Bio-Rad, Hercules, CA, USA). The membrane was blocked using 5% non-fat dried milk, 0.1% Triton X-100 in Phosphate Buffer Saline (PBST). The blot was incubated at 4 °C for 14 h with 5 ml of anti-ORF4 rabbit serum against recombinant 6xHis ORF4 (Cocalico Biologicals, Reamstown, PA, USA) diluted to 1:1000 in blocking solution. The membrane was washed three times with PBST and incubated at room temperature with anti-rabbit IgG, horseradish peroxidase-linked species-specific whole Ab from goat (Bio-Rad) diluted to 1:1000 in blocking solution for 2 h and examined using a chemical luminescence system (ECL Western blotting detection reagents; Bio-Rad) on a FluorochemQ imager (Protein Simple, Santa Clara, CA, USA).

### 4.3. In Vitro Transcription

Viral capped mRNA (P1, P6 and Sar55) was made from the linearized DNA (P1, P6 and Sar55) using the Promega Ribomax Large-Scale RNA Production System T7 (PRP 1300) and ARCA CAP TriLink biotechnologies (N-7003). A master mix using 10 µL T7 trans buffer 5×, 15 µL rNTP master mix, 5 µL T7 enzyme mix and ARCA CAP 0.4 µL for each sample was made. A total reaction volume of 50 µL consisting of linearized DNA template (4 µg) with 30 µL of master mix was prepared. Reactions were incubated at 37 °C in thermal cycler (Bio-Rad) for 4 h. The fidelity of transcripts was assessed and normalized by gel electrophoresis on 0.8% agarose gels containing ethidium bromide and visualized via UV illumination.

### 4.4. Transfection of ORF4+, ORF4– Huh7 Cells and Regular Huh7 S10-3 Cells

Huh7 liver cells are efficiently transfected, easy to passage and disperse better than PLC/PRF/5, HepG2/C3A or Caco-2 cells (25). RNA transfection was achieved using Mirus Trans-IT mRNA transfection kit. Transfection complex was made using Opti-MEM (300 µL, Invitrogen), trans IT boost reagent (9 µL), trans IT-mRNA (9 µL) and RNA (16 µL) for each plate (110 µL per seeded well), followed by incubation for 4 min at 37 °C in a humidified 5% CO_2_ atmosphere. After 48 h of transfection, cell passage was done (confluent cells were trypsinized with 0.5 mL trypsin and diluted 1:3 in medium and 0.5 mL was added to next 3 wells with 1 mL of media) and cells were left undisturbed for next 3 days and 5 days at 37 °C in a humidified 5% CO_2_ atmosphere. Similar experiments were repeated for checking the replication of gt1 Sar55 and gt3 P6 HEV RNA transcripts in the presence and absence of endoplasmic reticulum (ER) stressor compound (Tunicamycin 15 µg/mL).

### 4.5. Infection of HepG2/C3A Cells

HepG2/C3A cells are used for the HEV infection assay, since Huh7 cells cannot readily be infected. Three repeated freeze and thaw cycles were done with transfected cells after 5th day post-transfection (dpt) and 7th dpt. Cell lysates including both extracellular and intracellular virus were subjected to high-speed centrifugation step (10,000 rpm for 5 min) which separates the cell debris, and the supernatant was collected. HepG2/C3A cells were grown in 12 well plates to 70–80% confluency before infection with 800 µL supernatant to quantify virus infectivity. After 48 h, HepG2/C3A cells were passed 1:3 and 72 h later, they were harvested, fixed in 100% cold methanol, stained with rabbit anti truncated ORF2 antibody (primary) against HEV ORF2 and goat antirabbit IgG-PE, further followed by FACS analysis for the ORF2-positive cells quantification.

### 4.6. Flow Cytometry of In Vitro-Transcribed Capped HEV RNA Transfected (ORF4+ and ORF4−) Huh7 Cells, Huh7 S10-3 Cells, and Cellular Lysates Infected HepG2/C3A Cells

After 5 and 7 dpt, transfected cells were trypsinized using 0.25% trypsin and pelleted at 600× *g* for 5 min. Cells were then resuspended in 1 mL of 100% methanol and left at 4 °C for a minimum of 15 min. After an overnight storage at −80 °C, cells were spun out of methanol at 1500 rpm for 5 min, washed and resuspended in PBS. Cells were again pelleted, PBS removed and blocked-in blocking solution (10% Odyssey blocking buffer, 5% non-fat dried milk, 0.1% Triton X-100 in PBS) in 96-well plate for 30 min at 37 °C [44]. Cells were then washed with PBS once before probing with primary antibody—rabbit anti truncated ORF2 HEV diluted 1:200 in blocking solution for 30 min at 37 °C. After washing twice in PBS, cells were incubated with secondary antibody—goat anti-rabbit-phycoerythrin (PE) (Life Technologies) diluted to 1:400 PBS for 30 min at 37 °C. Cells were then washed twice in PBS, resuspended in 200 µL of PBS. Fluorescence was analyzed for 20,000 events using a flow cytometer (BD Accuri C6 Plus, Biosciences, San Diego, CA, USA). Gates were set to exclude dead cells based on forward and side scatter profiles and mock infected cells were used to gate background fluorescence. Similar protocol was performed for infected HepG2/C3A cells for the quantification of ORF2-positive cells after 5 dpt.

### 4.7. Indirect Immunofluorescence

At 5 and 7 dpt, transfected cells were fixed in 100% cold methanol and permeabilized with PBST. After blocking of non-specific binding with 5% non-fat milk (Sigma-Aldrich, St. Louis, MO, USA) in PBST for 60 min, immunostaining of ORF2-encoded capsid protein was performed. A 1:200 blocking buffer diluted rabbit anti truncated ORF2 HEV was added after 3 washes and PBS was subjected to cells for 5 min. A fluorescently labelled goat anti-rabbit IgG H&L antibody (Alexa Fluor 594; abcam, Cambridge, FL, USA) was used in a dilution of 1:400 in PBS to detect bound primary antibodies. DAPI was used to stain the nucleus. For quantification of virus infectivity, wells were manually observed for specific fluorescence, and the presence of fluorescent foci was recorded. A fluorescent focus was defined as a minimum of two cells showing clear intracytoplasmic fluorescence.

### 4.8. Statistics

GraphPad Prism Software 8.0 was used for data analysis using Student’s t-test to assess significance, as indicated in the figure legends. *p* < 0.05 was considered significant.

## Figures and Tables

**Figure 1 viruses-13-00075-f001:**
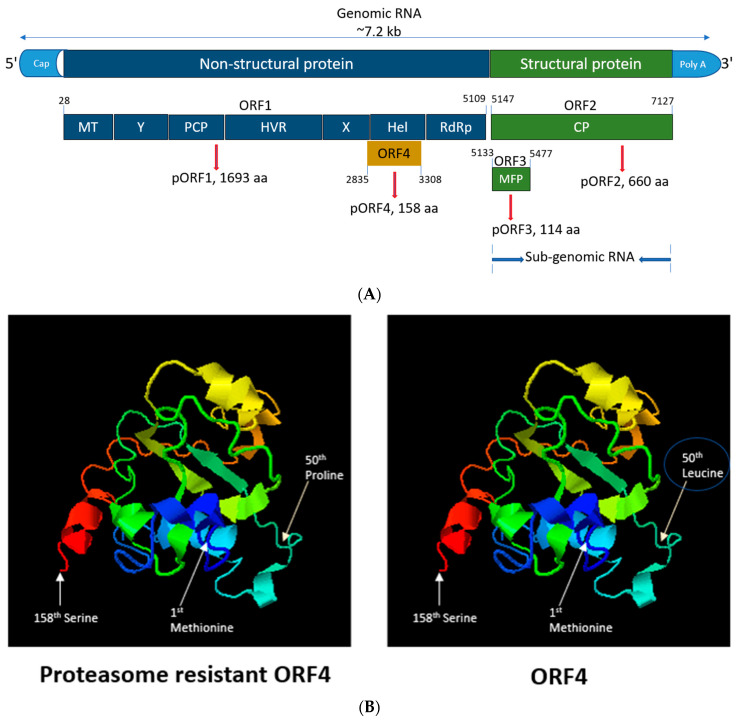
Genomic organization, functions of HEV ORFs and crystal structure of ORF4 protein. (**A**) The (+)-sense HEV genome is capped at the 5′ end, polyadenylated at the 3′ end and can be translated directly by host ribosomes [32]. ORF1 consists of various domains such as methyl transferase (MT), the X domain, papain-like cysteine protease (PCP), hypervariable region (HVR), the Y domain, helicase (Hel), RNA-dependent RNA polymerase (RdRp). Nucleotides 1–979 of HEV ORF1 encode the region responsible for capping the genome that gets translated to a protein of 110 kDa that possesses methyl transferase and guanylyltransferase activity. The protein allows for the transfer of the methyl group to guanosine triphosphate, giving m7-GTP, which gets covalently coupled to the 5′ end of the HEV genome [33]. The X domain binds poly-ADP ribose which is thought to play role in viral replication and/or translation. PCP is known for processing the HEV polyprotein into discrete function units. The Y domain possesses sequence conservation of several motifs within this region across all known HEV genotypes. HVR is rich in proline and plays a structural role as a flexible hinge between adjoining ORF1 regions [34]. Hel domain possesses NTPase activity and is capable of hydrolyzing ATP. RdRp synthesizes sense and antisense RNA transcripts using antisense and sense viral RNA templates, respectively [35]. A bicistronic subgenomic mRNA encodes both the ORF2 and ORF3 proteins of hepatitis E virus [36]. ORF2 encodes for the viral capsid protein (ORF2^C^) and the free form of the antigen (ORF2^S)^, and ORF3 encodes for multi-functional proteins, small phosphoproteins that is involved in viral morphogenesis, host signaling modification and egress [37,38]. ORF1 begins from 28 nucleotides and terminates at nucleotide position 5109. ORF1 translates into a non-structural protein (pORF1) with 1693 amino acids (aa). ORF2 starts at nucleotide 5147 till nucleotide position 7127. ORF2 translates into a structural protein (pORF2) with 660 aa. ORF3 starts at nucleotide 5133 and terminates at nucleotide 5477. ORF3 translates into structural protein (pORF3) with 114 aa. ORF4 starts at nucleotide 2835 and terminates at nucleotide 3308. ORF4 translates into pORF4 with 158 aa [19]. (**B**) Predicted 3D structure of the ORF4 protein using I-TASSER server [39,40] depicting the start (Methionine), end (Serine) and 50th aa mutation (from leucine to proline) leading to proteasome-resistant ORF4.

**Figure 2 viruses-13-00075-f002:**
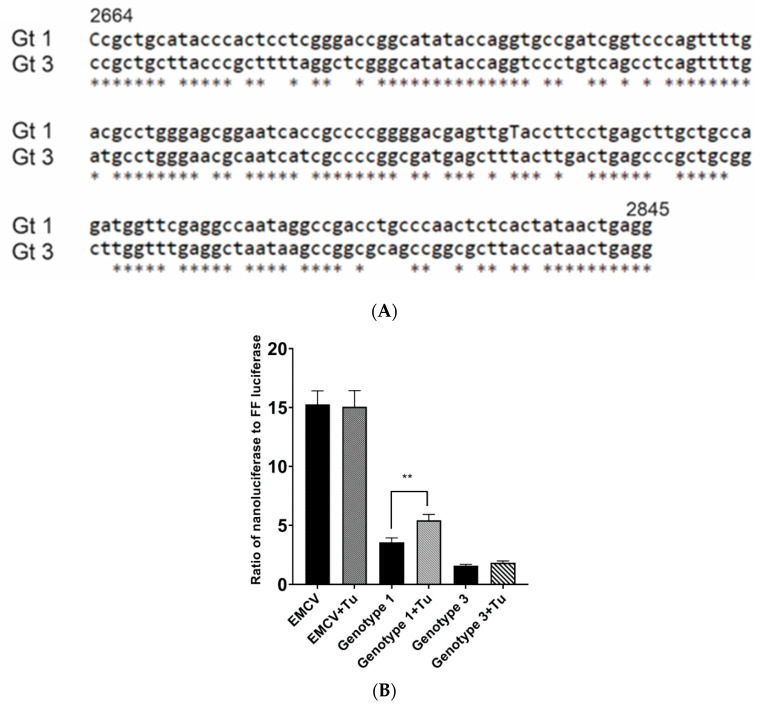
ORF4 within gt1 HEV regulated via IRES-like RNA that is upregulated through endoplasmic reticulum (ER) stress. (**A**) Sequence alignment of the gt1 (GenBank accession no. AF444002) region from 2664 to 2845 bp including IRES sequence compared to homologous sequence of gt3 (GenBank accession no. JQ679013) shows significant sequence homology. (**B**) IRES-like RNA activity is only seen in gt1 HEV. The ratio of nano luciferase (NL) to firefly luciferase (FFL) is compared in gt1 and gt3 IRES-like RNA element sequences in a dual luciferase reporter plasmid, in the presence of tunicamycin (ER stressor). (**C**) Translation maps of HEV ORF1 nucleotides encompassing the +1 frameshifted ORF4 reading frame start methionine from gt1 HEV and the comparable region from gt3. Methionine (blue arrow) the translation initiator amino acid of ORF4 within gt1 HEV is not present at the homologous position of the gt3 HEV. Each bar displays the mean of three independent biological experiments with four replicates per sample (** *p* < 0.01, *t*-test).

**Figure 3 viruses-13-00075-f003:**
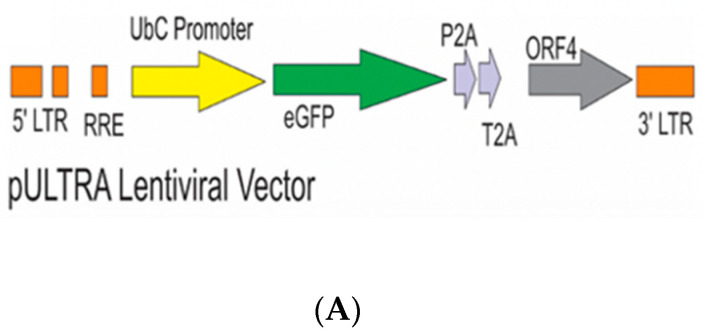
Lentiviral vector with ORF4 when transduced in Huh7 cell clones results in stable ORF4-expressing clones. (**A**) Schematic diagram of the lentiviral vector used with ORF4, GFP-tagged protein and promoter (T2A and P2A). (**B**) Workflow of lentiviral transduction in HEK293T cells. After 48 h of co-transfection using lipofectamine, cells were visualized for GFP expression. Supernatants were harvested and used to infect Huh7 S10-3 cells. Clonal cell lines were prepared using limiting dilution method. (**C**) Fluorescence microscope image illustrating GFP-positive cells which suggests the successful transduction of the lentiviral vector in both the cell lines (Empty = ORF4− and ORF4 = ORF4+). (**D**) Agarose gel electrophoresis after RT-PCR utilizing ORF4 specific PCR primers indicates the presence of ORF4 RNA and absence of ORF4 RNA in the transduced Huh7 ORF4+ and ORF4− cell clones, respectively. Each band represents ORF4 RNA of 500 bp. CL stands for clones that were formed after the limiting dilution. ORF4+ CL (3, 4, 5) represents the ORF4+ cell lines. ORF4−CL 4 represents the ORF4− cell line. (**E**) Western blots of Huh7 ORF4+ cell clones (3, 4, 5) and Huh7 ORF4− cell clone 1 lysates transiently expressing the HEV ORF4 protein of 20 kDa. The blot was probed with goat anti-rabbit IgG, horse radish peroxidase to detect the expression of the HEV ORF4 protein.

**Figure 4 viruses-13-00075-f004:**
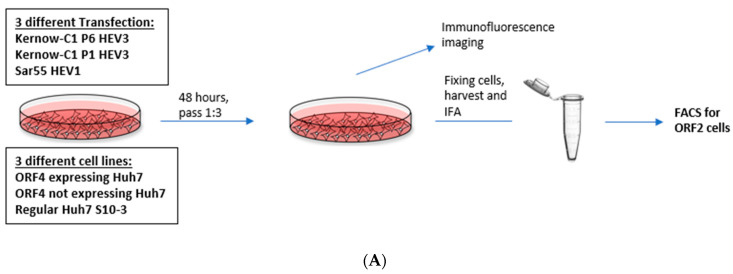
Presence of ORF4 from gt1 enhances the gt3 HEV and gt1 HEV replication. (**A**) Workflow of HEV transfection of target cells. ORF4+, ORF4− and Huh7 cell lines were transfected with in vitro-transcribed capped HEV mRNA (P1, P6 and Sar55). Each cell line was grown in DMEM with 10% heat-inactivated fetal bovine serum (HI-FBS). After 48 h, cells were passed 1:3. At 5 and 7 dpt, cells were harvested and used to detect HEV ORF2-positive cells. (**B**) Immunofluorescence detection of HEV ORF2 antigen in methanol-fixed ORF4+, ORF4−, Huh7 cell lines and respective mock cells after 5 dpt. Cells are stained with goat anti-rabbit IgG H&L combined with anti-rabbit Alexa fluor 594 (red), and 4′,6-diamidino-2-phenylindole (DAPI) (blue). Images serve to validate antibody staining specificity and are not necessarily representative images as most fields are devoid of antigen staining. (**C**,**D**) Flow cytometry quantification of ORF4+, ORF4− and Huh7 cells transfected with the capped RNA transcripts of the HEV gt3 (P1 and P6) strains and HEV gt1 (Sar55) strain, performed at 5 and 7 dpt. Samples were fixed in methanol and probed with rabbit anti-ORF2 followed by goat anti-rabbit PE antibodies. Each bar (mean ± SEM) represents separate transfections stained in parallel and display the mean of three independent biological experiments with nine replicates per sample (* *p* < 0.05, ** *p* < 0.01, and *** *p* < 0.001, **** *p* < 0.001 *t*-test). Statistically significant difference exists between ORF4+ cell lines transfected with HEV gt3 (P6, P1) and gt1 (Sar55) strains verses other cell lines respectively transfected with similar strains.

**Figure 5 viruses-13-00075-f005:**
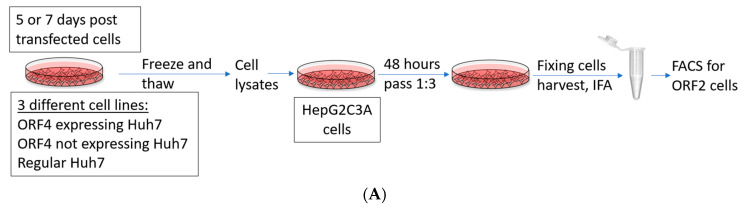
Presence of ORF4 from gt1 enhances gt3 HEV and gt1 HEV infectious titers. (**A**) Schematic representation of the workflow. HEV gt3 (P6 and P1) and gt1 (Sar55) transfected cells using duplicate cultures from the replication assay depicted in Figure 4 are lysed applying freeze and thaw cycles (3 times), centrifuged and supernatant harboring viral particles was collected and used to infect HepG2/C3A cells with equivalent lysate volumes. After 48 h, cells were passed 1:3 and incubated for 72 h. At 5 days post-infection (dpi), cells were harvested, fixed in 100% cold methanol, and quantified for ORF2-positive cells. (**B**,**C**) Flow cytometry quantification of HepG2/C3A cells infected with the supernatant from respective cell lysates (P6 ORF4, P1 ORF4, Sar55 ORF4, P6 empty, P1 empty, Sar55 empty, P6 Huh7, P1 Huh7 and Sar55 Huh7) performed at 5 dpi. Samples were fixed in cold methanol and probed with rabbit anti-ORF2 followed by goat anti-rabbit PE antibodies. Each bar (mean ± SEM) represents separate infections stained in parallel and display the mean of three independent biological experiments with nine replicates per sample (* *p* < 0.05, ** *p* < 0.01, and *** *p* < 0.001, *t*-test). Statistically significant difference exists between HepG2/C3A cell lines infected with equal volume of lysate from HEV gt3 (P6, P1) and gt1 (Sar55) strains from transfected cells expressing ORF4 verses infected cell lines not expressing ORF4.

**Figure 6 viruses-13-00075-f006:**
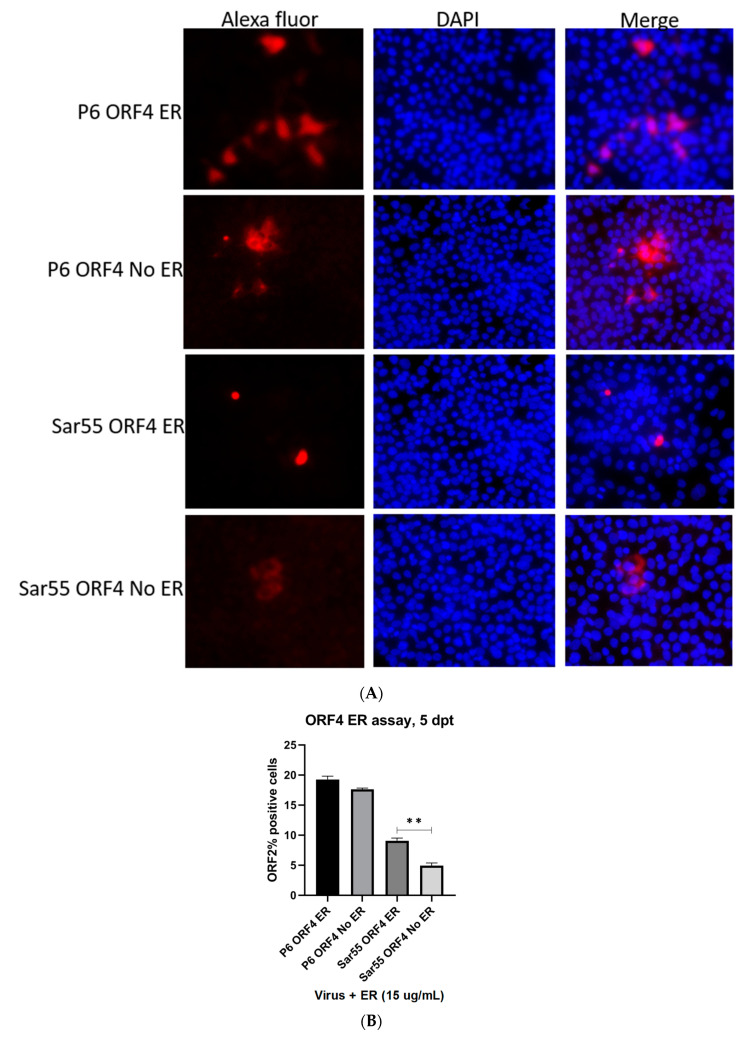
Sar55 replication enhanced in ORF4−expressing Huh7 cell lines treated with Tunicamycin. (**A**) Immunofluorescence detection of HEV ORF2 antigen in methanol fixed ORF4+ Huh7 cell treated with 15 µg/mL tunicamycin after 5 dpt with Alexa fluor 594 (red) and DAPI (blue). (**B**) Flow cytometry quantification of ORF4+ Huh7 cells transfected with the capped RNA transcripts of the HEV gt3 (P6) strain and HEV gt1 (Sar55) strain, performed at 5 dpt. Samples were fixed in methanol and probed with rabbit anti-ORF2 followed by goat anti-rabbit PE antibodies. Each bar (mean ± SEM) represents separate transfections stained in parallel and display the mean of three independent biological experiments with nine replicates per sample (** *p* < 0.01, *t*-test). Statistically significant difference exists between Sar55 transfected, endoplasmic stressor (tunicamycin) treated ORF4+ cell lines and untreated cell lines.

## Data Availability

Raw data not presented within the manuscript itself available on request.

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
