# Peer review of "Ectopic Expression of Genotype 1 Hepatitis E Virus ORF4 Increases Genotype 3 HEV Viral Replication in Cell Culture"

_viruses, 2021, doi:10.3390/v13010075_

Round 1
Reviewer 1 Report
Overview and general recommendation:
The authors describe an investigation of the effect of gt1 HEV ORF4 on gt 3 HEV. Although ORF4 does not exist in gt3 HEV, they showed that transiently expressing gt1 ORF4 increased the replication of both gt1 and gt3 HEVs in human hepatoma cells using multiple HEV infectious clones. Besides, they observed that replication of gt1 while not gt3 HEV is enhanced in ORF4 expressing cells with ER stressor, indicating IRES-like element activity is lacking in gt3 HEV. The findings may help to establish more efficient cell culture systems for HEV study. The overall topic of this study is very interesting and important, as knowledge regarding gt1 HEV ORF4 is largely scarce. The design and interpretation of the study are in general well performed, but the presentation needs to be improved. I ask the authors to specifically address each of my comments in their response.
Major comments
- The authors conducted in silico analyses and stated that IRES-like RNA element and ORF4 are not existed in gt3 HEV, what about gt2 and gt 4 HEVs? Also, there are so far approximately 80 gt1 HEV genomic sequences in the GenBank, how conserved is the ORF4 in gt1 HEV?
- I don’t think Fig.1B and 1C are essential to be presented. Otherwise, please supplement further in-depth description in the maintext. Is there any structural difference between ORF4_50Pro and Proteasome resistant ORF4_50Leu in Fig.1B? Letters in Fig.1C are too small to read.
- For the confirmation of ORF4+ Huh7 cells, the authors showed RT-PCR bands in Fig.2D. However, how could the contamination of plasmid residues be excluded? I did not see a DNase digestion step in the Method section.
- The immunofluorescence assays figures are not presented very well. Firstly, the resolutions are not high enough. Secondly, in Fig.3B, it seems like all cells are positive for anti-ORF2 staining in P6 and P1 gt3, which is impossible. Thirdly, in Fig.6A, is it necessary to divide fluorescence into three channels?
- In the infectivity assay, please specify the viral loads used for HepG2/C3A cells infection. This point is very important, in the replication assay, there are already more viruses in P6 and Sar55 ORF4+ Huh7 cells. I am wondering why HEV is more infectious from ORF4+ Huh7 cells lysate, please just concisely explain it.
- Reference citations need to be checked carefully.
Minor comments
- Introduction, line 42: citation of reference “1” is unnecessary.
- Introduction, line 54: citation of references “16-18” is incorrect.
- Introduction, line 61: “RdRp” appears the first time, please define the abbreviation.
- Introduction, lines 66 and 68: please unify “in vitro” and “in vitro” in the maintext.
- Results, line 88: the GenBank accession no. “AAF444002” is incorrect and the reference citation of “12” is improper.
- Results, line 90: Fig.2A should not appear in advance of Fig.1B and 1C.
- Discussion, lines 178-181: supporting reference is lacking.
- Discussion, lines 184-185: this interpretation is too strong which needs to be softened.
- Discussion, lines 246 and 250: citations of references “46” and “47” are identical.
- Materials and methods, line 311: please correct the GenBank accession no.
- References, line 555: the “e” should be capitalized.
- Figure 1A: the order of X domain and Y domain in HEV genome is incorrect.
- Figure 2, line 458: please correct the GenBank accession no., which should be “AF444002”.
- Figure 2A: I suggest to use “Gt” instead of “Gen”.
- Figure 2C: what do nucleotide numbers at the bottom indicate? From 2800 to 2870?
- Figure 5A: letters are too small to read.
- Figure 6B: please change “orf4” at the bottom to “ORF4”.
Author Response
We greatly appreciate the thorough review. We have made many revisions based on this valuable feedback. Please see the attached point by point reply to the critiques.

Reviewer 2 Report
The authors describe a new way to enhance HEV replication in vitro either of gt1 and gt3 strains.
The experimental conditions are well described and clearly showed.
My only minor concern is about the discussion section. I guess it must be reduced giving the right citations to the reader that wants to go deeper in the arguments
Author Response
We greatly appreciate the favorable review. If there are specific references that the reviewer feels would benefit the discussion or specific discussion paragraphs they would like to see omitted from the draft we will be happy to comply.
